# Exploring Relations between Cloud Morphology, Cloud Phase, and Cloud Radiative Properties in Southern Ocean Stratocumulus Clouds

Jessica Danker[1], Odran Sourdeval[2], Isabel L. McCoy[3,4], Robert Wood[5], and Anna Possner[1]

[1]Institute for Atmospheric and Environmental Sciences, Goethe University Frankfurt, Frankfurt, Germany
[2]Univ. Lille, CNRS, UMR 8518 - LOA - Laboratoire d'Optique Atmosphérique, F-59000 Lille, France
[3]Cooperative Programs for the Advancement of Earth System Science, University Corporation for Atmospheric Research, Boulder, Colorado, USA
[4]Rosenstiel School of Marine and Atmospheric Science, University of Miami, Miami, Florida, USA
[5]Atmospheric Sciences, University of Washington, Seattle, WA, USA

**Correspondence:** Jessica Danker (danker@iau.uni-frankfurt.de)

**Abstract.** Marine stratocumuli are the most dominant cloud type by area coverage in the Southern Ocean (SO). They can be divided into different self-organized cellular morphological regimes known as open and closed mesoscale-cellular convective (MCC) clouds. Open and closed cells are the two most frequent types of organizational regimes in the SO. Using the liDAR-raDAR (DARDAR) version 2 retrievals, we quantify 59 % of all MCC clouds in this region as mixed-phase clouds (MPCs) during a 4-year time period from 2007 to 2010. The net radiative effect of SO MCC clouds is governed by changes in cloud albedo. Both cloud morphology and phase, have previously been shown to impact cloud albedo individually, but their interactions and their combined impact on cloud albedo remain unclear.

Here, we investigate the relationships between cloud phase, organizational patterns, and their differences regarding their cloud radiative properties in the SO. The mixed-phase fraction, which is defined as the number of MPCs divided by the sum of MPC and supercooled liquid cloud (SLC) pixels, of all MCC clouds at a given cloud-top temperature (CTT) varies considerably between austral summer and winter. We further find that seasonal changes in cloud phase at a given CTT across all latitudes are largely independent of cloud morphology and are thus seemingly constrained by other external factors. Overall, our results show a stronger dependence of cloud phase on cloud-top height (CTH) than CTT for clouds below 2.5 km in altitude.

Preconditioning through ice-phase processes in MPCs has been observed to accelerate individual closed to open cell transitions in extratropical stratocumuli. The hypothesis of preconditioning has been further substantiated in large-eddy simulations of open and closed MPCs. In this study, we do not find preconditioning to primarily impact climatological SO cloud morphology statistics. Meanwhile, in-cloud albedo analysis reveals stronger changes in open and closed cell albedo in SLCs than MPCs. In particular few optically thick (cloud optical thickness > 10) open cell stratocumuli are characterized as ice-free SLCs. These differences in in-cloud albedo are found to alter the cloud radiative effect in the SO by 21 W m$^{-2}$ to 39 W m$^{-2}$ depending on season and cloud phase.

## 1 Introduction

In the Southern Ocean (SO), marine stratiform low clouds cover between 40 % to 60 % of the ocean surface (Wood, 2015) and due to their high albedo, they play a key role in the radiative balance of the Earth (Randall et al., 1984; Ramanathan
et al., 1989; Hartmann et al., 1992; Chen et al., 2000). Especially at high latitudes, many marine stratocumuli occur as mixed-phase clouds (MPCs). In contrast to pure liquid clouds, MPCs contain a mixture of supercooled liquid and ice. The phase partitioning between liquid and ice in stratocumuli strongly impacts the cloud radiative properties (Sun and Shine, 1994; Matus and L'Ecuyer, 2017; Korolev et al., 2017). Due to the complex microphysics in MPCs, our understanding of the impact of phase partitioning on the radiative properties of these low-level clouds remains limited (McCoy et al., 2015; Tan and Storelvmo,
2019). Furthermore, the cloud phase feedback remains poorly represented in models, particularly in the SO (Bony et al., 2006; Zelinka et al., 2012, 2013), which represents a critical region to compute climate sensitivity (Gettelman et al., 2019; Zelinka et al., 2020). Given the extensive coverage of MPCs in the SO and their impacts on cloud albedo, it is especially important to observe, understand, and quantify the cloud radiative properties of MPCs in the SO.

Stratocumuli are divided into different self-organized morphological regimes referred to as open and closed mesoscale-
35 cellular convective (MCC) clouds which are associated with different cloud fractions (Atkinson and Zhang, 1996; Wood and Hartmann, 2006). In the SO, open and closed cells are the two most frequent types of MCC clouds (Muhlbauer et al., 2014). Especially in austral winter, open MCC reach their highest occurrence frequency whereas closed MCC occur more often in summer. Due to their organizational differences, the cloud fraction of closed MCC clouds is on average about 30 % higher than for open MCC clouds (Wood and Hartmann, 2006) and thus closed MCC clouds reflect more incoming shortwave radiation.
Moreover, McCoy et al. (2017) showed that even for the same cloud fraction closed MCC clouds have a higher cloud albedo than open MCC clouds. Therefore, it is important to understand the processes which are related to the occurrence of the two types of MCC clouds in low-level clouds and their transition to quantify their radiative effects on Earth's climate. One process controlling the shift from closed to open cell convection is the formation of precipitation (Feingold et al., 2010) through a decoupling of the boundary layer induced by precipitation (Abel et al., 2017). Further, Yamaguchi and Feingold (2015) find
that not only the formation of precipitation but its spatial extent is essential for the transition of the MCC clouds regimes. Even large-scale meteorological events like marine cold air outbreaks which are often found in the SO (Fletcher et al., 2016a) can impact the occurrence of MCC clouds. McCoy et al. (2017) show that open MCC clouds preferentially form during marine cold air outbreaks.

The potential link between cloud phase and cloud field organization in mixed-phase stratocumuli was first explored by Abel
et al. (2017), Eirund et al. (2019a), and Tornow et al. (2021). Abel et al. (2017) analyze aircraft observations in the North Atlantic and find that the transition from mixed-phase closed to open MCC clouds is accompanied by a shift from supercooled dominated MPCs to more glaciated MPCs. They further show that the key factor for the onset of closed to open cell transition is precipitation. Tornow et al. (2021) address the question of which ice processes are relevant for precipitation during the

transition. They introduce the notion of *preconditioning* whereby efficient riming-related processes lead to more favorable conditions for cloud breakup and accelerated the transition of an overcast stratocumulus deck into a broken cloud field. Eirund et al. (2019a) demonstrate in a case study of Arctic stratocumulus that mixed-phase open MCC clouds have a larger cell size than pure liquid open cells.

All three studies utilize field observations of particular situations together with numerical models allowing them to disentangle the potential impact of different processes in greater detail. Yet, it remains to be seen whether preconditioning due to ice-phase processes occurs often and widely enough to impact statistics of cloud morphology and cloud albedo.

In this study, we investigate the connections between cloud phase, cloud organization, and cloud albedo in the SO. In order to investigate these connections for a wide range of cases, we use active satellite data of the Afternoon Constellation (A-Train) from 2007 to 2010. The cloud phase is analyzed by using a vertically integrated cloud phase classification which we describe in Sect. 2.2. Moreover, we focus on the seasonal changes during austral winter (June–August, JJA) and summer (December–February, DJF) because these seasons have the highest occurrence frequency of open and closed MCC clouds, respectively. In Sect. 3.1, we analyze the quality of our cloud phase classification and investigate the link between cloud phase, season, and cloud morphology. In Sect. 3.2, we analyze the connections between freezing behavior and cloud phase under different seasonal or morphological conditions. We examine the dependence of cloud top temperature (CTT) and cloud top height (CTH) in open MCC clouds, closed MCC clouds, and low-level clouds. Finally, we address the question of how cloud phase and cloud morphology impact cloud albedo (Sect. 3.3).

## 2 Data and Methods

### 2.1 DARDAR and MODIS

The raDAR-liDAR (DARDAR) v2 data product (Delanoë and Hogan, 2010; Ceccaldi et al., 2013) combines data from the Cloud-Aerosol Lidar and Infrared Pathfinder Satellite Observations (CALIPSO) and CloudSat satellites. The two products are collocated onto the CloudSat footprints (∼1.1 km). The advantage of combing lidar and radar measurements is that due to their different wavelengths they detect different parts of the hydrometer spectrum. While the lidar is sensitive to small particles and thus small liquid droplets, the radar is dominated by larger particles and thus mainly by ice particles. In this study, DARDAR v2 is used which significantly reduces the overestimation of supercooled pixels in the lowest part of the troposphere compared to DARDAR v1 (Ceccaldi et al., 2013). We analyze data covering the time period from 2007 to 2010 and focus on the SO (40° S to 65° S). While Huang et al. (2021b) report large differences in cloud phase detection between various satellite products, which struggle specifically with MPCs, they use the DARDAR v1 which is known to overestimate supercooled liquid. In contrast, DARDAR v2 is validated with several ground-based measurements in the Antarctic by Listowski et al. (2019) who also show that DARDAR v2 has the ability to capture the seasonal cycle of supercooled liquid cloud fraction. Nevertheless, MPCs with very low ice crystal number concentrations which are common in the SO might still be misidentified as supercooled liquid. Further, we chose the DARDAR product as it merges information from two active instruments and thus provides a vertically resolved cloud phase in contrast to passive satellites which only resolve cloud phase at cloud top. The

DARDAR cloud classification additionally requires a temperature profile only in the radar mask and the strong lidar backscatter layers ($\beta_{532} > 2 \times 10^{-5}\,\mathrm{m^{-1}\,sr^{-1}}$) of the DARDAR classification algorithm for further details see Ceccaldi et al. (2013). The temperature and other thermodynamic variables like sea surface temperature (SST) and surface wind speeds are collocated on the CloudSat track by the European Center for Medium-Range Weather Forecasts (ECMWF)-AUX. Moreover in this study, we combine the DARDAR v2 product with the Moderate Resolution Imaging Spectroradiometer (MODIS) cloud product (MYD06_L2) Collection 6 (C6) version from the Aqua satellite (Platnick et al., 2015). The liquid water path (LWP) and the cloud optical thickness (COT) are provided by MODIS. Further, we derive the in-cloud albedo ($\mathrm{Alb}_{cld}$) from the MODIS COT to remain consistent with DARDAR's horizontal pixel resolution of 1.1 km. Following Berner et al. (2015) based on Platnick and Twomey (1994), we use the equation:

$$\mathrm{Alb}_{cld} = \frac{(1-g)\tau}{2 + (1-g)\tau} \tag{1}$$

Here, COT is indicated as $\tau$ and the asymmetry parameter is g = 0.85 which assumes small water droplets. McFarquhar and Cober (2004) find that MPCs peak at g = 0.85 and liquid clouds at g = 0.87. Further, Gayet et al. (2002) show that in MPCs the asymmetry parameter ranges from 0.82 to 0.85 which is similar to values in liquid clouds. They find higher values of g are typically found in liquid clouds with high liquid water content whereas lower values of g (0.73–0.80) are found in ice clouds. This corresponds to findings by Shcherbakov et al. (2005) and Xu et al. (2022) who demonstrate that the asymmetry parameter is g = 0.77 in cirrus clouds in the SH. As differences between liquid clouds and MPCs are similar the asymmetry parameter g = 0.85 is used for both liquid and MPCs.

## 2.2   Vertically Integrated Cloud Phase

To analyze the cloud phase, we use the DARDAR cloud classification, which provides a vertically resolved cloud phase with a 60 m resolution from surface to 25.08 km. This vertically resolved cloud phase is based on a lidar and radar mask provided by the DARDAR algorithm (for details see Tab. 1 of Ceccaldi et al. (2013)). Therefore, when the lidar is extinguished the DARDAR classification can only determine the layer to be ice cloud, warm rain, or cold rain. The DARDAR classification has 17 different categories which are displayed in the example tracks of DARDAR in Fig. 1 a and S4. In this study, the following categories of DARDAR are grouped into four categories: (1) *Ice* (ice clouds, spherical or 2D ice, and highly concentrated ice), (2) *Sup* (supercooled water and multiple scattering due to supercooled water), (3) *Mix* (supercooled + ice), and (4) *Liq* liquid warm. To reduce the vertical cloud phase into a vertically integrated cloud phase, we first identify the highest and lowest cloud levels which are categorized as *Sup*, *Mix*, or *Liq*. The height of the highest cloud level is defined as the cloud top height (CTH) and the lowest as the liquid cloud base height (CBH). Thereby excluding pure ice clouds which we exclude as the MCC algorithm is based on the LWP (see Sect. 2.3) . As we are only interested in low-level clouds, any data point with a CTH above 3 km is excluded from this analysis. The surface cluttering of the radar can cause noise up to 2 km which can not be clearly distinguished from the signal, especially at heights below 720 m and thus clouds are missed (Marchand et al., 2008). Even though some studies (Liu et al., 2012; Fletcher et al., 2016b) consider anything roughly below 1 km as ground clutter, Mioche et al. (2015) show that in comparison with ground-based observation the cloud fraction of DARDAR is 10 % lower from 500 m

to 1000 m while in the range from 0 m to 500 m it is 25 % lower. Thus, in this study, we consider 720 m as the threshold for surface clutter similar to other studies (Kay and Gettelman, 2009; Huang et al., 2017; Noh et al., 2019; Listowski et al., 2019). In order to correctly identify the cloud phase, however, we require one level below the liquid CBH. Thus, we restrict to only clouds with a liquid CBH at 780 m or above. Moreover, we remove any multi-layer clouds, defined here as clouds with three or more consecutive vertical levels marked as clear or fillvalues. As the constructed vertical resolution of DARDAR is 60 m, three levels equal a distance of 240 m which is also the oversampled vertical resolution of CloudSat (effective vertical resolution 480 m). Thus, this distance ensures that multi-layer clouds are two separated clouds with a sufficiently large separation.

In order to assign one cloud phase to a certain data point in DARDAR, we need to reduce the DARDAR cloud classification in the vertical dimension. Therefore, all data points are classified into MPCs, liquid clouds, or clear depending on their vertical phase distribution (Fig. 1 b). Here, we only analyze pixels. Liquid clouds are considered to be clouds that only consist of *Liq*, *Sup*, or *Sup* above *Liq* ($Sup \rightarrow Liq$). As MPCs, we consider five different types: only *Mix*, *Mix* above *Ice* ($Mix \rightarrow Ice$), *Sup* above *Ice* ($Sup \rightarrow Ice$), any combination of *Sup* and *Mix* ($Sup \leftrightarrow Mix$), and any combination of *Sup* and *Mix* above *Ice* ($Sup \leftrightarrow Mix \rightarrow Ice$).

Typically, the lidar in our cloudy pixels extinguishes within 300 m (five vertical levels) (interquartile range = 360 m – 240 m) and thus provides information beyond the cloud top phase. As mentioned above, the radar mask of the DARDAR classification requires the ECMWF wet bulb temperature to distinguish between ice ($\leq 0\,°C$), and liquid ($> 0\,°C$) or rain ($> 0\,°C$) phase. Therefore, this could lead to uncertainty in the cloud phase classification close to $0\,°C$ especially if the lidar is extinguished. As this affects cloud phase classification at temperatures close to $0\,°C$, this should not lead to a bias in the overall cloud phase distinction. Furthermore, for temperatures below $0\,°C$, the radar classification cannot distinguish between supercooled drizzle and ice. In particular in the SO, supercooled drizzle is observed in stratocumulus clouds at temperatures near -10 °C (Mace and Protat, 2018). Further, Silber et al. (2019) show that at the observation station McMurdo, Antarctica supercooled drizzle can persist at temperatures below -25 °C for several hours. While it might be possible that the *Mix* classification of DARDAR itself is affected as this category is supercooled liquid from lidar and ice from radar. We find it unlikely that multiple layers of *Mix* could be affected as the lidar would extinguish in the presence of drizzle and the vertical lidar resolution of CALIPSO is 30 m. As most MPCs that contain *Mix* have mixed layers with a thickness of roughly 480 m (eight vertical levels in DARDAR) (see Fig. 1a and S4), the MPCs with identified *Mix* levels by the radar retrieval are unlikely to be pure drizzle. However, the misclassification of supercooled drizzle as *Ice* could lead to false identification in MPCs when the lidar is extinguished especially in the cloud category $Sup \rightarrow Ice$, as the *Ice* in these clouds could be supercooled drizzle. Supercooled drizzle is reported to be misclassified as ice by several studies (Cober and Isaac, 2012; Zhang et al., 2017, 2018; Villanueva et al., 2021) in particular at temperatures above -10 °C.

To further test the uncertainties of misclassified supercooled drizzle, we checked how our results are changed if only clouds with an effective radius of $0\,\mu m < R_e < 14\,\mu m$ at cloud top are investigated. Thus, precipitating clouds should be excluded as $R_e > 14\,\mu m$ at cloud top initiates drizzle (Han et al., 1995; Rangno and Hobbs, 2005; Rosenfeld et al., 2012; Freud and Rosenfeld, 2012). However, we only find slight changes with this threshold (compare Fig.3 with Fig. S2). Further, as MODIS

is not able to calculate $R_e$ in over 50 % of the identified MPCs (Fig. S1) and as we would also exclude correctly identified

precipitating MPCs the threshold of $R_e$ is not used as a constraint in this study.

The ECMWF cloud top temperature (CTT) is defined as the temperature from ECMWF at CTH. As shown in four examples in Fig. S5, our data set, which is combined with MODIS, also provides the CTT from MODIS. However, we decide to use the ECMWF CTT for two reasons: 1) because it will be more consistent with the DARDAR classification methodology which is also based on the ECMWF temperature and further because CTH between DARDAR and MODIS varies and 2) because the

MODIS CTT exhibits unrealistically large and abrupt changes of more than 10 °C within a distance of 2 km (Fig. S5). From a brief visual inspection, it seems to be related to jumps in MODIS CTH which are not detected by the active satellites. Further, we find that the MODIS CTH is often higher than that of DARDAR.

## 2.3 MCC Classification

The MCC regime identifications are developed by applying the supervised neural network algorithm designed in Wood and

Hartmann (2006) to collection 6.1 MODIS Aqua LWP swath data. The algorithm uses the power density function and power spectrum of LWP to determine whether swath sub-scenes (256 km x 256 km areas) fall into one of three categories: open MCC, closed MCC, or cellular but disorganized. See Eastman et al. (2021) for more information on the collection 6.1 MCC identifications. To collocate the MCC data set with the CloudSat track, the haversine distance for all DARDAR data points to the middle of each MCC scene is calculated. The MCC regime of the nearest MCC scene within a radius of 128 km is set for

each DARDAR data point.

## 3 Results

### 3.1 Stratocumulus Climatology

Cloud morphology and reflectivity are vertically integrated quantities of a two-dimensional cloud field. In order to explore the links between morphology, phase, and their combined potential relation to cloud albedo, a vertically integrated catego-

rization for cloud phase was built (Fig. 1) as described in Sect. 2.2. Here, we address the quality and limits of our vertically integrated cloud phase and their seasonal differences. Further, the possible connections between cloud phase and organization are investigated.

According to our cloud phase classifications, most MPCs are characterized by a *Mix* cloud layer with ice-phase precipitation below cloud base in the SO. Whereas commonly in the SO, many MPCs are described to consist of a supercooled liquid top

with ice precipitation below in satellites studies (e.g. Hu et al., 2010; Morrison et al., 2011; Huang et al., 2012; Ahn et al., 2018; Mace et al., 2021) and also by some ground-based and in situ measurements (e.g. Shupe et al., 2008; Niu et al., 2008; D'Alessandro et al., 2021; McFarquhar et al., 2021). Note that spaceborne studies can either be based on passive instruments which typically only cover the cloud top phase (Morrison et al., 2011) or also include active instruments like lidar or radar (Hu et al., 2010; Huang et al., 2012; Ahn et al., 2018; Mace et al., 2021) which can penetrate layers below cloud top. Recently,

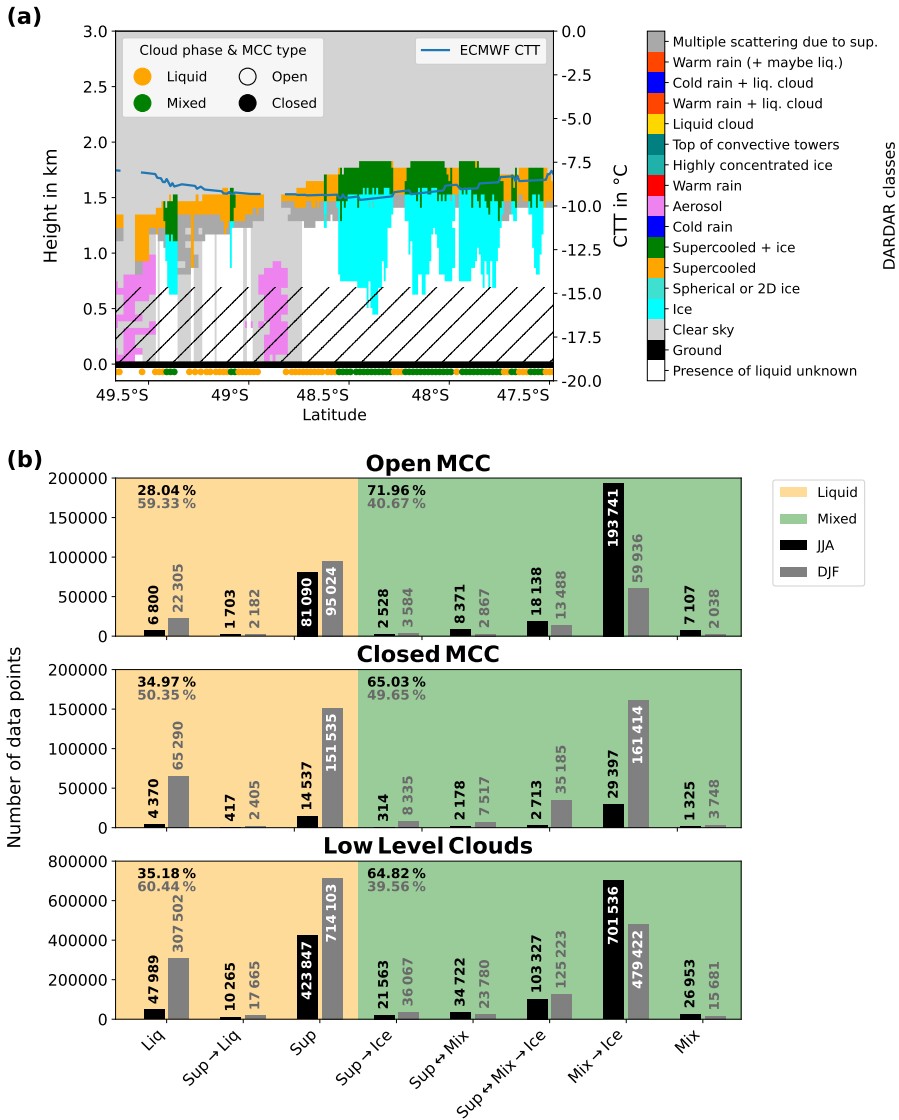

**Figure 1.** (a) Example track of the DARDAR categorization on December 1, 2007. The hatched area displays levels below 720 m. The colored circles below the ground show our vertically integrated cloud phase classification and the MCC type for every second data point. (b) Histogram of data points of vertically integrated cloud phase subcategories divided into liquid clouds (orange) and MPCs (green) for JJA (black) and DJF (grey) from 2007 to 2010. Overall percentage of liquid clouds and MPCs is indicated in each panel separately for JJA (black) and DJF (grey). "→" indicates layer on top of the next one. "↔" indicates interchangeable layers.

the comparison of active satellites from CALIPSO or CloudSat with ground-based or in situ measurements shows that their products underestimate the occurrence of MPCs in the SO (Ahn et al., 2018; Mace et al., 2021). This is further supported by many field campaign studies which observe the presence of ice in the supercooled top layer even at relatively high CTT

($> -5\,°C$) in MPCs (e.g. Huang et al., 2017; Ahn et al., 2017; Lang et al., 2021; Zaremba et al., 2021). The previous version of DARDAR (v1) shows a tendency to detect too many liquid or supercooled liquid pixels in the lower troposphere (Ceccaldi et al., 2013). As the study by Huang et al. (2012) uses the DARDAR v1 product they find more supercooled liquid-topped MPCs which is likely due to the bias in the DARDAR v1 cloud classification algorithm.

While most of our MPCs (more than 95 %) contain a *Mix* layer that is determined by both the radar (ice) and the lidar (supercooled liquid), we also include *Sup* over *Ice* clouds in our MPC classification. This category is the most uncertain as the phase distinction between ice and rain is solely based on the wet bulb temperature (frozen $< 0\,°C$) once the lidar has saturated and only radar retrieval is available (Delanoë and Hogan, 2008; Ceccaldi et al., 2013). Thus, these clouds could also be pure supercooled liquid clouds with or without freezing rain below cloud base (see Sect. 2.2). The impact of this possible misclassification only marginally affects our MPC classification, as most MPCs contain a *Mix* cloud layer and further excluding them did not substantially alter our results.

Of our liquid low-level clouds, about 90 % in austral winter and 60 % in summer are supercooled at cloud top and almost all of them (99 %) belong to the *Sup* category and thus remain supercooled at cloud base in the SO. We identify almost no low-level liquid clouds that show DARDAR rain categories below cloud base ($\sim 0.5\,\%$). However, the DARDAR algorithm was not primarily designed to detect precipitation and as the CloudSat radar is contaminated by surface clutter, only heavy and moderate drizzle can be detected at heights below roughly 720 m and 860 m, respectively (Marchand et al., 2008). Thus, for many liquid low-level clouds which have a CTH of around 1.2 km (Fig. 2 g–h) light drizzle rates at cloud base could have been missed at lidar saturation which explains the too low drizzle rates. If we define precipitating clouds as clouds with an effective radius of $R_e > 14\,\mu m$ then roughly 10 % and 3 % of low-level liquid clouds are precipitating in winter and summer, respectively (Fig. S1 a and c). These values are in agreement with Mülmenstädt et al. (2015) who show that the rain probability of liquid clouds at all levels is roughly 10 % at $45°\,S$ and 3 % at $60°\,S$. Though they use DARDAR v1 to identify the cloud phase, they use the 2C-PRECIP-COLUMN product based on retrievals from CloudSat to calculate rain probability. Further, this latitudinal gradient in precipitation is also reported by Mace et al. (2021) who investigate MPCs with satellite and ground-based measurements. They also show that about 33 % of MPCs in the SO produce supercooled precipitation. We find similar values of 30 % (winter) and 40 % (summer) of MPCs that have $R_e > 14\,\mu m$ (Fig. S1 a and c). Additionally, we find that on average liquid low-level clouds are 57 % optically thinner than their mixed-phase counterparts calculated independent of season and thus are more unlikely to contain sufficient water content to generate precipitation. As most liquid clouds are optically thin and are not precipitating especially in summer, this could either hint towards a mixed-phase detection bias in DARDAR, which we find unlikely as discussed above or suggest that most optically thicker supercooled liquid clouds generate ice and become MPCs. Interestingly, liquid (and supercooled liquid) closed MCC clouds are optically thicker than open and low-level clouds. This could indicate a potential link between cloud phase and cloud morphology, however, as discussed below we find no further evidence for this link.

To further investigate the quality of the cloud phase classification, we also examine the CTT range. Figures 2 a–c display the probability density functions (PDFs) of CTT, which are normalized individually for each cloud phase and season. The normalization is also performed separately within all panels. In low-level clouds, the CTT range spans from $-30\,°C$ to $15\,°C$

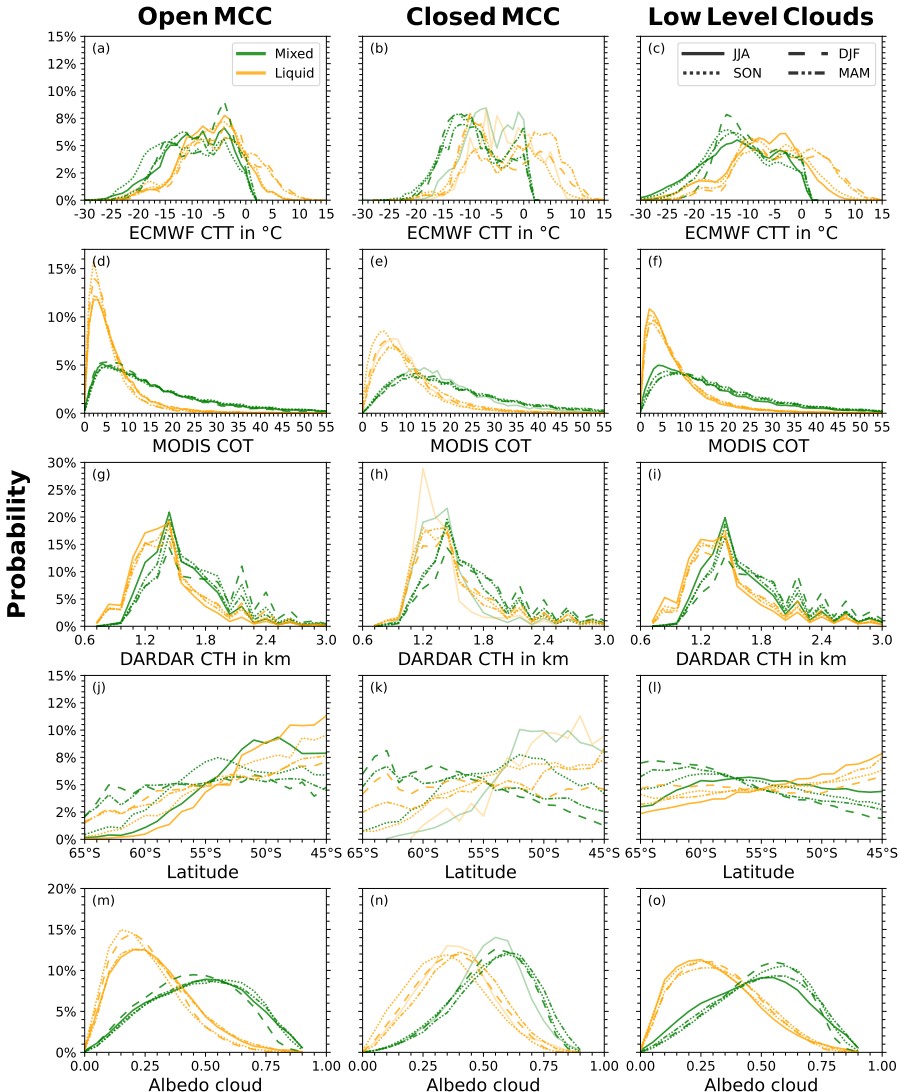

**Figure 2.** Seasonal probability density functions (PDFs) of (a–c) CTT, (d–f) COT, (g–i) CTH, (j–l) latitude and (m–o) cloud albedo for (left) open MCC, (middle) closed MCC, and (right) low-level clouds with bin width of 1 °C, 1, 120 m, 1° and 0.05, respectively. The PDFs are normalized for each cloud regime type, phase and season individually. In JJA only 5.1 % of the annual closed MCC clouds occur and therefore closed MCC in JJA are indicated by more transparent color shading.

in liquid clouds and from -30 °C to 3 °C in MPCs in the SO (Fig. 2 a–c). We note, that the reason for the occurrence of MPCs above 0 °C is related to the fact that in the radar mask of the DARDAR v2 algorithm the wet-bulb temperature of 0 °C is used as a threshold (Delanoë and Hogan, 2010; Ceccaldi et al., 2013). Seasonal changes in the CTT range are mainly found in the maximum temperature of liquid clouds above 0 °C. These temperature ranges of MPCs and liquid clouds are in agreement with

other satellite studies of the SO (Morrison et al., 2011; Mason et al., 2014). The low-level MPCs occur most often at around -15 °C. This peak corresponds to the temperature of the growth habit of dendritic ice crystals and secondary ice processes from ice-ice collisional break-up (Riley and Mapes, 2009; Mignani et al., 2019).

Overall, we observe a seasonal shift from predominantly MPCs (∼65 %) during austral winter to predominantly liquid clouds (∼60 %) during austral summer (Fig. 1). Listowski et al. (2019) also use the DARDAR v2 product and exhibit in their Fig. 8 that during both austral winter and summer low-level liquid clouds occur more often than MPCs in the SO. However, in their analysis, they include low-level clouds in the range of surface cluttering which leads to limitations in identifying ice at those heights and thus could lead to a bias towards liquid clouds. If we visually confine the analysis of Listowski et al. (2019)

to heights above 780 m in austral winter, the occurrence of MPCs is more pronounced. Thus, our findings are consistent with them if clouds with higher uncertainty in cloud phase distinction are excluded.

In addition to the seasonal cycle in cloud phase, we observe a seasonal cycle in MCC regime. As previous studies show, the predominant MCC regime shifts from open cell MCC during austral winter to closed cell MCC during summer (Muhlbauer et al., 2014; McCoy et al., 2017). Open cell MCC is found relatively homogeneously across the year with the lowest rate

of occurrence in austral summer (16 %) and the highest rate of occurrence in winter (25.4 %). Meanwhile closed cell MCC display a strong seasonal shift. McCoy et al. (2017) explained this seasonal shift in MCC occurrence with the varying strength and frequency of occurrence of marine cold air outbreaks. Merely 5.1 % of all closed cell MCC are found in austral winter while 40.5 % of all closed cell MCC occur in summer. This results in fewer than 100 clouds per 1 °C CTT bin in some bins in austral winter. Thus, if the austral winter closed MCC clouds are further subdivided by other variables e.g. CTT, CTH, or Lat,

their climatology might not yield sufficient data points for a reliable statistical analysis which is indicated by more transparent colors in that panel or season in our figures (Fig. 2 b,e,h,k,n, 3, 4, and 5).

In general, both MCC regimes exhibit a similar CTT distribution (Fig. 2 a and b). Mixed-phase MCC clouds feature one peak at around -4 °C in all seasons which is especially strong in austral summer and a second peak at roughly -15 °C which is more pronounced in closed cells. The first peak falls in the temperature range (-3 °C to -8 °C) of secondary ice production

by the Hallet-Mossop process (Hallett and Mossop, 1974). While the second peak at -15 °C is found in many ice formation studies (Magono, 1962; Takahashi et al., 1995; Libbrecht, 2005; Mignani et al., 2019; Sullivan et al., 2018; Silber et al., 2021b), multiple ice processes can occur at this temperature range. This second peak will be extensively discussed in Sect. 3.2. We note that in MCC clouds the CTT range only extends down to about -20 °C to -25 °C which is likely caused by the condition of the MCC algorithm that cloud tops need to be within 30 °C of the surface temperature (McCoy et al., 2017). Further, we only

identify small cloud phase seasonal changes in open and closed MCC clouds compared to the overall low-level cloud statistic (Fig. 2 c). During austral winter we see slightly more open MPCs than low-level MPCs and during austral summer more closed MPCs.

We observe that the seasonal decrease in cloud occurrence south of 60° S is stronger in MPCs than in liquid clouds (Fig. 2 l). This is consistent with Listowski et al. (2019), who also find that the occurrence of MPCs is reduced to a larger degree than

that of liquid clouds. This behavior is likely related to seasonal differences in sea ice extent (not shown). This connection between the sea ice edge and low-level cloud fraction is also found in other studies (Taylor et al., 2015; Wall et al., 2017;

Morrison et al., 2018). Further, the latitudinal difference in cloud organization shows that in the open cell regime the decrease of cloud occurrence in both MPCs and liquid clouds is more substantial than in low-level clouds (Fig. 2 j). This might also be impacted by the sea ice extent as open MCC clouds are correlated with marine cold air outbreaks (McCoy et al., 2017) which shift equatorward in austral winter along with the sea ice edge. During austral winter we observe a detection limit in the MCC regimes south of 60° S, as the algorithm is based on the passive MODIS Aqua satellite instrument which depends on solar insolation for measurements. However, we do not find it likely that this limit is impacting our hypothesis as the reduction of cloud occurrence at latitudes closer to the pole also appears in austral spring which is not impacted by this detection limit.

Overall, we are confident that our cloud phase classification of MPCs contains ice and that we can therefore trust our phase classification. Further, the climatology of SO stratocumuli as characterized by DARDAR v2 did not display any evidence that organization and cloud phase are interlinked in the full climatology. Although, we observe that closed cells remain in the SLC regime at higher COT than observed for open cell and low-level clouds.

## 3.2   Link of Freezing Behavior and Cloud Phase

In this section, we analyze whether different predictors of ice occurrence in stratocumuli display a varied behavior in differently organized clouds. From these analyses, we can determine whether there are statistical relationships that suggest that individual freezing processes vary in their effectiveness in clouds characterized by different cloud dynamics.

Here, we analyze the cloud phase fraction between MPCs and supercooled liquid clouds (SLCs) Their cloud phase fractions (mixed fraction and supercooled liquid fraction) are defined as the number of MPC or SLC pixels divided by their sum. The cloud phase dependence on CTT has already been studied by several other publications to find a relationship between ice formation and CTT (e.g. Bühl et al., 2013; Zhang et al., 2014, 2015; Silber et al., 2021a, b). Thus, we restrict our analysis for the rest of this study to a CTT range from -20 °C to 0 °C. We choose this temperature range as most clouds in the open and closed MCC regime have CTTs above -20 °C. This restriction does not affect the overall distribution of MPCs and liquid clouds except for the fact at we remove all pure liquid clouds (Fig. S1). Therefore, this analysis is restricted to MPCs and clouds containing a supercooled liquid layer or only supercooled liquid, referred to as SLCs.

Overall, the mixed fraction is much higher in austral winter at the same CTT than in summer for all three investigated cloud regimes (Fig. 3). This seasonal increase in mixed fraction during austral winter could either be caused dynamically or due to increased INP availability at colder temperatures. An increase in surface fluxes or higher surface wind speeds in austral winter could indicate a dynamic reason. However, we did not find substantial seasonal changes in either SST or surface wind speed (Table 1). As described in Sect. 3.1 we find an equatorward shift of MPCs in austral winter (Fig. 2, Table 1). However, we find that the mixed fraction at the same CTT is independent of latitude (Fig. 4). Further, we find that the same CTTs are reached at lower CTHs in austral winter than in summer (Fig. S3 and S4). Thus, we hypothesize that the vertical distribution of INP might influence the seasonal difference in mixed fractions. McCluskey et al. (2019) investigate the simulated vertical INP distribution based on observational data from the Clouds, Aerosols, Precipitation, Radiation, and atmospherIc Composition Over the southeRN ocean campaign (CAPRICORN) and show that independent of the season the INP concentration in the SO is higher closer to the surface as the main source of INP is sea spray aerosols (Burrows et al., 2013; DeMott et al., 2016; Vergara-

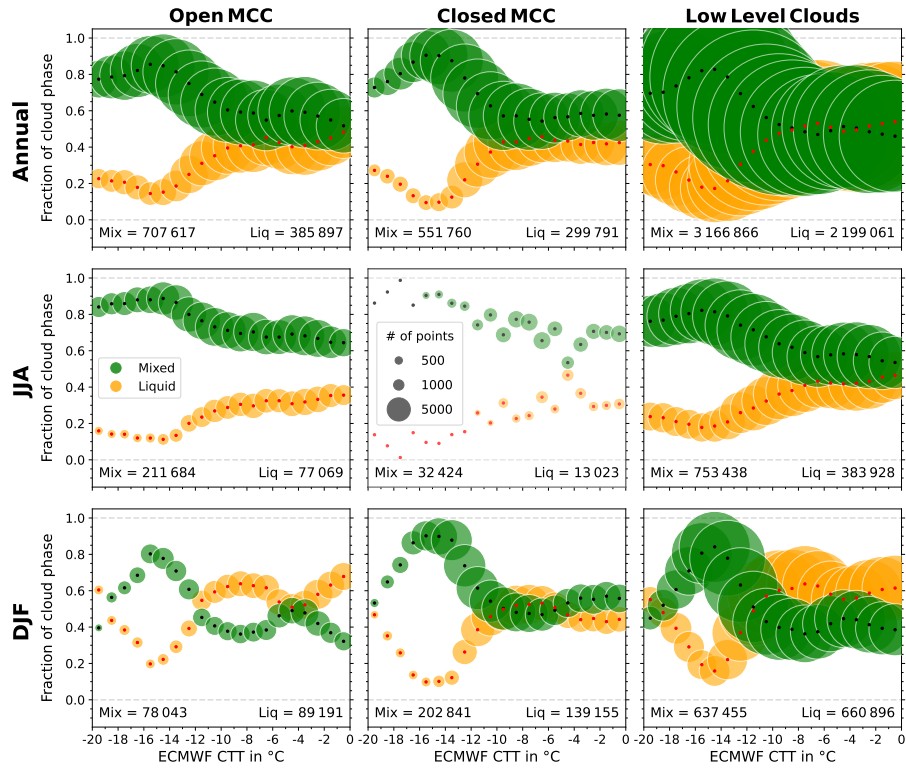

**Figure 3.** Supercooled liquid and mixed fraction binned by CTT from -20 °C to 0 °C with a bin width of 1 °C (2007 - 2010) for (top) all seasons, (middle) austral winter and (bottom) austral summer in (left) open MCC, (middle) closed MCC, and (right) low-level clouds. As only 5.1 % of the annual closed MCC clouds occur in JJA the panel is displayed in more transparent color shading.

Temprado et al., 2017; Huang et al., 2021a). Further, Fig. 4 of McCluskey et al. (2019) displays that the INP concentration is slightly lower (∼ 35 % at the surface, ∼ 55 % at around 3 km) at all heights in austral winter than in summer. Nonetheless, we find a higher mixed fraction in MCC and all low clouds for CTTs above -12 °C at CTHs between 1.4 km and 2.3 km which decreases with higher CTHs (Fig. S4). Surprisingly, this behavior is not observed during austral summer. Therefore, we suggest
that the increase in mixed fraction in austral winter is related to the higher mixed fraction at CTHs between 1.4 km and 2.3 km. But it remains unclear what is causing this effect as higher INP concentration closer to the surface is also found in austral summer (McCluskey et al., 2019) which does not show a higher mixed fraction at lower CTHs.

In austral summer, the mixed fraction remains below 0.5 for temperatures higher than -12 °C with a secondary peak at around -5 °C in open MCC and all low-level clouds. Surprisingly, this peak is not observed in closed MCC clouds. This could
potentially be related to a detection bias close to 0 °C as the mixed fraction at temperatures above -10 °C is lower for clouds with 0 μm < $R_e$ < 14 μm (Fig. S2). However, even for clouds with 0 μm < $R_e$ < 14 μm this secondary peak is barely detectable and much weaker than in open MCC and all low-level clouds. This secondary peak in open MCC and all low-level clouds in the mixed fraction occurs at temperatures at which the secondary ice production by the Hallet-Mossop process is especially active

**Table 1.** Geometric mean and standard deviation factor of different cloud properties during austral winter and summer in the CTT range from -20 °C to 0 °C. The mean values are calculated separately for open MCC, closed MCC and low-level clouds which are further subdivided into MPCs and SLCs. [The geometric standard deviation factor is shown in brackets and should be interpreted as a range from "geomean/geostd" to "geomean*geostd"].

| | | CTH in km | Lat in ° S | SST in K | Wind in m s$^{-1}$ | LWP in g m$^{-2}$ | COT | Alb$_{cld}$ |
|---|---|---|---|---|---|---|---|---|
| **JJA** | | | | | | | | |
| **Open** | **MPC** | 1.57 (1.22) | 48.17 (1.10) | 279.7 (1.01) | 11.14 (1.45) | 151.5 (2.29) | 13.19 (2.52) | 0.45 (1.64) |
| | **SLC** | 1.41 (1.24) | 47.15 (1.10) | 280.6 (1.01) | 10.52 (1.51) | 47.0 (2.12) | 5.38 (2.26) | 0.27 (1.75) |
| **Closed** | **MPC** | 1.49 (1.25) | 48.38 (1.10) | 278.6 (1.01) | 9.10 (1.56) | 164.8 (1.89) | 15.98 (1.97) | 0.52 (1.41) |
| | **SLC** | 1.43 (1.27) | 48.34 (1.10) | 279.0 (1.01) | 7.78 (1.67) | 63.1 (1.95) | 8.02 (1.96) | 0.36 (1.55) |
| **Low-Level** | **MPC** | 1.60 (1.25) | 51.44 (1.13) | 277.8 (1.02) | 10.49 (1.55) | 146.4 (2.29) | 12.93 (2.53) | 0.45 (1.66) |
| | **SLC** | 1.43 (1.29) | 49.66 (1.13) | 279.0 (1.02) | 9.46 (1.66) | 48.7 (2.21) | 5.81 (2.34) | 0.28 (1.77) |
| **DJF** | | | | | | | | |
| **Open** | **MPC** | 1.78 (1.27) | 52.68 (1.13) | 279.5 (1.01) | 10.76 (1.48) | 95.8 (2.65) | 11.22 (2.30) | 0.42 (1.61) |
| | **SLC** | 1.56 (1.30) | 51.72 (1.12) | 279.9 (1.01) | 10.18 (1.48) | 27.0 (2.36) | 4.52 (2.08) | 0.24 (1.72) |
| **Closed** | **MPC** | 1.79 (1.28) | 56.60 (1.10) | 276.2 (1.01) | 9.50 (1.60) | 136.7 (2.21) | 16.59 (1.99) | 0.52 (1.41) |
| | **SLC** | 1.64 (1.28) | 56.14 (1.11) | 276.2 (1.01) | 8.63 (1.61) | 52.3 (2.28) | 8.73 (2.05) | 0.37 (1.58) |
| **Low-Level** | **MPC** | 1.84 (1.28) | 56.21 (1.12) | 276.6 (1.01) | 9.08 (1.65) | 114.7 (2.52) | 14.12 (2.18) | 0.48 (1.53) |
| | **SLC** | 1.61 (1.31) | 55.08 (1.12) | 277.2 (1.02) | 8.36 (1.68) | 37.7 (2.52) | 6.50 (2.23) | 0.31 (1.73) |

(Hallett and Mossop, 1974). A recent study by Silber et al. (2021b) in the Arctic also shows that the liquid water occurrence in clouds reduces at roughly -6 °C and -15 °C. They conclude that this is caused by a more efficient vapor growth of ice at these temperatures. Moreover, their second minimum at -15 °C corresponds to the strong increase in the mixed fraction from -12 °C to -16 °C that we find for all cloud regimes in austral summer and the annual mean. This increase occurs across all latitudes in the SO (Fig. 4) and is also seen in austral winter, though due to the overall higher mixed fraction in winter, the increase is not as pronounced. This peak in ice formation at roughly -15 °C is found in several studies (Magono, 1962; Takahashi et al., 1995; Libbrecht, 2005; Mignani et al., 2019; Sullivan et al., 2018; Silber et al., 2021b), though, there are different reasons for this increase in the numbers of ice crystals. Takahashi et al. (1995) find that ice-ice collisional break-up (secondary ice formation) favors this temperature range at roughly -15 °C. Further, Mignani et al. (2019) investigate whether an ice crystal that grows at temperatures between -12 °C and -17 °C forms due to primary or secondary ice formation. They find that only every eighth ice crystal contains an INP and thus that secondary ice formation is more important at this temperature range. Another possible

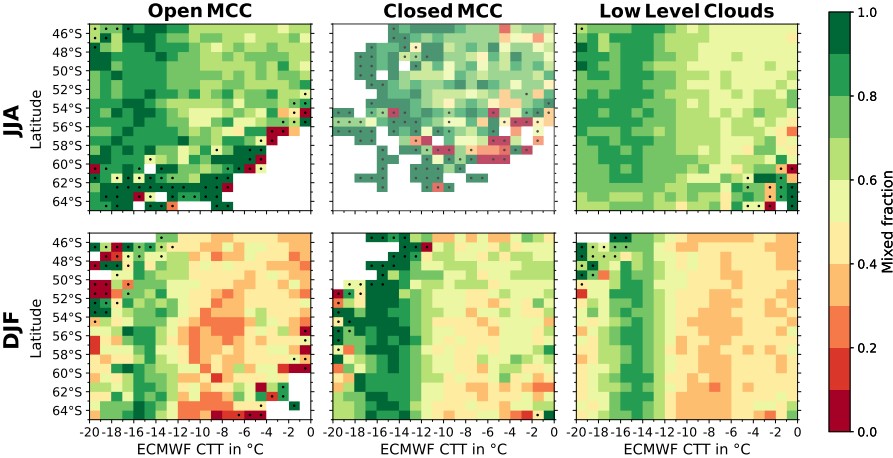

**Figure 4.** Two-dimensional histograms of mixed fraction against CTT and latitude for (left) open MCCs, (middle) closed MCC, and (right) low-level clouds in austral (top) winter and (bottom) summer. Dotted bins indicate bins with less than 50 data points. As only 5.1 % of the annual closed MCC clouds occur in JJA the panel is displayed in more transparent color shading.

way of ice formation at this temperature range would be droplet shattering. However, a modeling study by Sullivan et al. (2018) shows that droplet shattering seems to play only a minor role for clouds with a cloud base temperature below 12 °C (285 K) as the droplets cannot grow to a sufficient size to shatter. As our data set does not include INP information, we cannot determine which ice processes are causing the mixed fraction at -15 °C to increase.

At temperatures below -16 °C, there is a strong decrease in mixed fraction in all cloud regimes during austral summer which is less pronounced in the annual mean and austral winter. We suggest that the cause for the reduction in mixed fraction is due to a rapid glaciation of MPCs at these temperatures due to updraft or moisture limitation. A strong increase in fully glaciated clouds at these temperatures is found by D'Alessandro et al. (2021) who base their study on data from the Southern Ocean Clouds, Radiation, Aerosol Transport Experimental Study (SOCRATES) and cover the time period from 15 January to 28 February 2018. Figure 4 of D'Alessandro et al. (2021) shows that at roughly -17 °C the relative occurrence frequency of MPCs and SLCs decreases along with temperature, whereas the frequency of ice clouds increases rapidly at this temperature. Further, a direct comparison of SOCRATES flight observations from D'Alessandro et al. (2021) with our mixed fraction in Fig. S7 shows a similar trend across the CTT range in low-level clouds during January and February, though our mixed fraction shows higher values than the in-cloud flight measurements. Thus, this supports the rapid glaciation of MPCs at temperatures below -16 °C as soon as ice is formed via the Wegener-Bergeron-Findeisen process. D'Alessandro et al. (2021) suggest that this is caused by the activation of INP at these temperatures. On the other hand, we find it unlikely that CTH-dependent INP limitation is the primary cause for the decrease in mixed fraction below -16 °C at high CTHs as it seems to be unaffected by CTH (Fig. S4). Our analysis shows that the mixed fraction during austral winter is not decreasing as strongly as in summer. As a temperature-dependent activation of INP should not change with season this cannot fully explain the seasonal differences

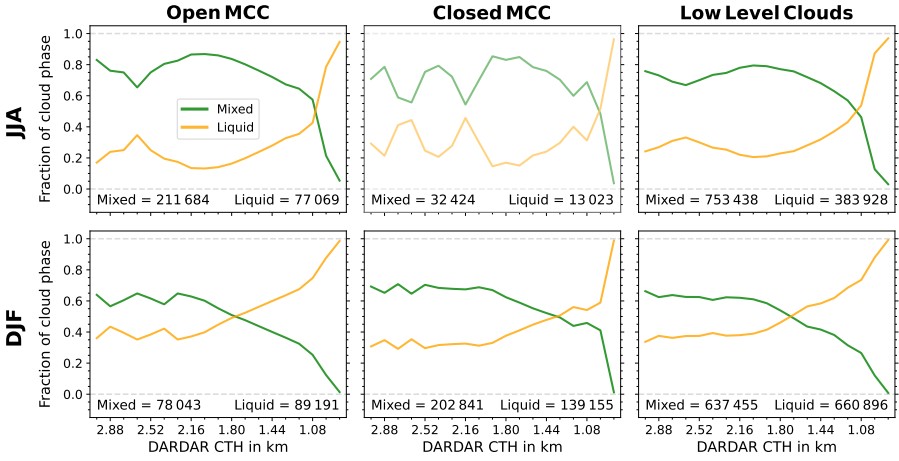

**Figure 5.** Supercooled liquid and mixed fraction binned by CTH from 0.78 km to 3 km with a bin width of 0.12 km (2007 - 2010) during austral (top) winter and (bottom) summer for (left) open MCC, (middle) closed MCC, and (right) low-level clouds. As only 5.1 % of the annual closed MCC clouds occur in JJA the panel is displayed in more transparent color shading.

we observe. Therefore, we do not think this is a result of different INP activation within these clouds, but we propose that the
supercooled liquid water is depleted due to an increased decoupling of the marine boundary layer.

In agreement with the findings of Sect. 3.1, the mixed fraction of closed MCC clouds is higher than that of open MCC clouds. Abel et al. (2017) show that transitions between closed and open cells in the Northern Hemisphere extratropics may be driven by precipitation as opposed to a pure boundary layer deepening. This idea is further supported by findings from Tornow et al. (2021) who introduce the idea of preconditioning by ice-phase processes, which accelerate the precipitation-driven transition.
Early onset of precipitation by riming processes and subsequent sublimation trigger an earlier boundary layer decoupling and preconditions the boundary layer for an earlier transition. If preconditioning would be a dominant process in stabilizing the sub-cloud layer and forcing closed to open transitions, one would expect this to manifest in phase statistics across the two morphological regimes. However, mixed fraction curves (Fig. 3) and phase statistics show little changes with respect to cloud morphology. Furthermore, any differences detected are small in comparison to seasonal changes in cloud phase, which are
driven by other factors than mesoscale organization. Thus, a prevalence of open MCC clouds towards MPCs, which would be consistent with accelerated transitions from closed to open MCC clouds through precipitation is not found.

We also investigate the dependence of mixed-phase occurrence upon CTH. Typically, the cloud depth is a better indicator for thermodynamic or dynamic changes in the boundary layer or radiative changes in stratocumulus clouds than the CTH (Wood et al., 2008; Bretherton, 2015). However, even though we derive a liquid CBH to reduce the contamination of surface clutter
from the radar, this CBH is highly biased in the distance from CTH because the lidar will be completely extinguished by clouds with a COT greater than 3.5 (Delanoë and Hogan, 2008). Thus, the geometrical cloud depth would also be biased as most clouds have a COT greater than 3.5 (Fig. 2 d–f). Nevertheless, the CTH might still give some insight to surface forcing and the mixing strength in the boundary layer (Bretherton et al., 2010).

In general, we observe that the mixed fraction increases with CTH from roughly 0 to around 0.6 to 0.8 in all cloud regimes and during both austral winter and summer (Fig. 5). We find seasonal differences in the height at which the mixed fraction surpasses the supercooled liquid fraction. This height is lower during austral winter. As clouds with CTHs below 1 km can only have a small vertical extent, this could potentially lead to a bias towards SLC occurrence at CTHs below 1 km as thicker clouds tend to form ice as discussed in Sect. 3.1. However, this is the same for all seasons and cloud morphologies. Thus, differences across seasons and between open and closed cells can still be interpreted. Further, we show that MPCs appear at higher CTHs than SLCs in all cloud regimes (Table 1). This is in agreement with a field campaign study in the Arctic that shows that MPCs tend to have higher CTHs than SLCs (Achtert et al., 2020). The mean CTHs between open and closed MCC clouds are similar during austral summer, whereas during austral winter at least for MPCs we see higher CTHs in open cells. Many studies show that there are CTH differences between the two morphological regimes with higher CTHs in open MCC clouds (Muhlbauer et al., 2014; Glassmeier and Feingold, 2017; Jensen et al., 2021). A study using ground-based and satellite observations in the Eastern North Atlantic shows that closed MCC clouds have a lower mean CTH (Jensen et al., 2021). Further, Glassmeier and Feingold (2017) demonstrate in a large-eddy simulation that open cells favor deeper boundary layer heights and thus also higher CTHs. In global data, Muhlbauer et al. (2014) reveal that the mean CTH in open MCC clouds is about 100 m higher than in closed cells which is similar to what we see in MPCs in austral winter. However, they also investigate the mean CTH in SO which did not show a substantial mean CTH difference between open and closed cells.

Deeper boundary layers associated with higher CTHs are often decoupled and favor conditional instabilities associated with stronger vertical updrafts which in turn favor ice growth and potentially ice formation through secondary ice processes. This is shown by a SOCRATES study from Wang et al. (2020) who investigate generating cells in the SO and show that within these generating cell updrafts ice particles occur more often and are also larger than outside. Thus, this favors ice precipitation inside the updraft cores. Further, they still find substantial amounts of ice outside the generating cells which suggests that turbulent mixing in the boundary layer is important to reduce differences between inside and outside of the updrafts. The stronger precipitation within updrafts is also confirmed by large eddy simulations (e.g. Keeler et al., 2016; Zhou et al., 2018; Young et al., 2018; Eirund et al., 2019b). The updraft strength can also vary depending on the organizational regime. Wood et al. (2011) analyze the updraft strength in MCC regimes in a case study over the Southeast Pacific and show that while open cells can reach higher updraft velocities, closed cells also exhibit moderate updrafts. Apart from the updrafts, the CTH and MPC occurrence also depends on the sources of mixing in the stratocumulus-topped boundary layer. Therefore, we test for indicators of surface-generated turbulence such as SST and $\Delta T$ (difference between SST and 2 m air temperature). However, neither variable displayed the expected trend (not shown). Thus, if there is a correlation between ice occurrence and vertical acceleration it does not seem to be driven by surface fluxes (Fig. S6). We cannot evaluate the importance of cloud top generated turbulence and cloud scale overturning circulations for CTHs in SO stratocumuli due to data limitations. However, Lang et al. (2022) show that cloud top generated mixing especially in closed MCC is affecting the occurrence frequency during the diurnal cycle. Further, they find that wind shear due to the relatively large climatological near-surface winds in the SO may also be a stronger generator of boundary layer turbulence than in other regions. Overall, this could suggest that the mechanisms of mixing (turbulence and circulation) may play a larger role in CTH than previously thought (McCoy et al., 2017).

In summary, our analysis shows that across regimes of varied subsidence, clouds that form in likely decoupled layers requiring moderate updraft cloud cores to be maintained, are more likely to sustain ice formation in mixed-phase stratocumuli. Our analysis of the different freezing behavior across cloud morphologies further supports our climatological findings which show that the sustained ice formation in MPC stratocumuli does not primarily depend on cloud morphology but is constrained by other environmental factors.

## 3.3 Relationship between Cloud Phase, Cloud Morphology, and Cloud Albedo

Here, we examine how cloud phase and cloud morphology may change the cloud albedo in the SO. The cloud albedo physically depends on the LWP and cloud droplet number concentration (in liquid clouds). Variations of cloud phase, cloud fraction, and different organizational regimes can alter the LWP and the cloud droplet number concentration and hence, impact cloud albedo and COT. For the same total water content, liquid clouds typically have a higher cloud albedo than ice clouds, because liquid water droplets are smaller than ice crystals, and thus reflect more incoming solar radiation due to their greater surface area. Thus, the cloud albedo in MPCs varies depending on the phase partitioning of supercooled liquid and ice (McCoy et al., 2014a, b). Further, any optically thick cloud (COT > 10) typically contains ice, which suggests that clouds with a substantial LWP can sustain ice formation. Consistently, we find that the LWP and COT of MPCs are much higher than those of SLCs independent of organizational regime and season (Table 1). This is in agreement with other studies, which also show that supercooled liquid layers in MPCs are much thicker than in pure (supercooled) liquid clouds (Shupe et al., 2006; Achtert et al., 2020). In austral winter, both mixed-phase MCC clouds have a similar LWP. Whereas in austral summer, open MPCs have a lower LWP than mixed-phase closed cells. We should note that the MODIS LWP algorithm used here does not distinguish between MPCs and liquid clouds and retrieves the LWP as based on a liquid cloud. Therefore, the LWP in MPCs is likely overestimating the true LWP. This can lead to an overestimated LWP of about 15 % for stratiform MPCs (Khanal and Wang, 2018).

Not only the cloud phase influences the cloud albedo but also the cloud fraction and cloud morphology. Loeb et al. (2007) determine that the variability of all-sky albedo from the Clouds and the Earth's Radiant Energy System (CERES) is dominantly controlled by variations in cloud fraction. The cloud fraction of closed MCC regimes is typically higher than in open MCC regimes (Muhlbauer et al., 2014). Moreover, McCoy et al. (2017) investigate differences in the cloud fraction albedo relation between open and closed MCC clouds and show that in general closed MCC clouds have a higher albedo. Additionally, they exhibit that even for the same cloud fraction the cloud albedo of closed MCC clouds is about 0.05 higher on average than the albedo of open MCC clouds. Our analysis of in-cloud albedo confirms their findings that closed cells are more reflective than open cells (Table 1). In addition, we also see that in-cloud albedo differences between closed and open cells are even stronger in SLCs (JJA: 0.09, DJF: 0.13) compared to MPCs (JJA: 0.07, DJF: 0.10). This is caused by stronger differences between optically thin (COT < 10) open and closed cells in SLCs compared to MPCs. Whereas in open cells roughly 80 % of the SLCs are optically thin, in closed cells about 45 % have COT values larger than 10. Differences in in-cloud albedo ranging between 0.07 to 0.13 correspond to a cloud-radiative effect of $21\,\mathrm{W\,m^{-2}}$ to $39\,\mathrm{W\,m^{-2}}$ when assuming typical solar insolation of $300\,\mathrm{W\,m^{-2}}$ in the SO. Thus, a reduction in ice-phase occurrence in a warming climate is likely to impact open cell clouds

more strongly than closed cell clouds changes in clouds with larger optical depth have a weaker impact on cloud scene albedo. Further, we find a considerable seasonal change in in-cloud albedo in open cells, which is not observed in closed cells. This is
even stronger in open MPCs than in SLCs. This seasonal decrease of the in-cloud albedo in open clouds is correlated with a strong decrease in LWP from austral winter to summer.

## 4   Discussion and Conclusions

So far only a few studies have investigated the potential link between cloud organization and cloud phase in stratocumuli (Abel et al., 2017; Eirund et al., 2019a; Tornow et al., 2021). All of them are based on field campaigns in the Northern Hemisphere
which observe particular cases and extensively analyze their processes with numerical models. Thus, in this study, we explore whether this link between cloud phase and morphology can also be found in SO cloud statistics obtained from spaceborne lidar-radar retrievals.

An advantage of using remote sensing data is that they cover a broad variety of cases and have almost global coverage. The spatial coverage of passive satellites would be even greater than that of active satellites. However, the cloud phase in
passive instruments can only be evaluated at cloud top and often show a supercooled layer there (e.g. Hu et al., 2010). Thus, passive satellite retrievals potentially miss many MPCs which form ice below the detected supercooled layer. To partially circumvent this issue, we use active instruments to determine the cloud phase. An important part of this study is to test the quality of our cloud classification. In agreement with previous studies, our vertically integrated cloud phase classification based on the DARDAR v2 cloud classification seems to provide a good representation of SO MPCs as compared to previous
assessments (Huang et al., 2017; Ahn et al., 2017; Lang et al., 2021; Zaremba et al., 2021). The greatest uncertainty in MPC classification is introduced by the *Sup* over *Ice* subcategory as ice and rain are classified merely based on temperature once the lidar is extinguished. Thus, some of these clouds could be SLCs with supercooled rain below cloud base, instead of ice. However, as most MPCs classified in this study include a *Mix* layer in their vertical composition which can only be determined if both, lidar and radar retrievals are available simultaneously, the majority ($> 95\,\%$) of all classified MPCs are not subject to
this potential misclassification. However, our cloud statistic may not be representative of all clouds, as especially in austral winter many shallow clouds form with cloud tops below 780 m. However, a remote sensing-based phase classification of these very low clouds from above is not possible due to rapid saturation of the lidar within the liquid layer and surface clutter issues with the radar. However, the seasonal cycle of MCC regimes when imposing this restriction is similar to that of the full SO climatology (Muhlbauer et al., 2014; McCoy et al., 2017) and thus, missing the very low clouds ($< 780\,\text{m}$) should not influence
our conclusions regarding the link of cloud phase and organization.

We find that all optically thick low-level clouds tend to generate ice formation, as all detected liquid clouds and SLCs are mostly ($> 80\,\%$) optically thin (COT $< 10$). We, therefore, hypothesize that any optically thicker supercooled cloud provides a favorable environment for ice occurrences which leads to a phase conversion from SLCs to MPCs. Although we do not find any evidence for a potential link between cloud phase and cloud morphology in the full climatology, we observe that closed
cells remain in the SLC regime at higher COT than observed for open cell and low-level clouds.

The observed relationship between phase occurrence and CTT, suggests that while CTT may be a strong functional relationship for the nucleation rate of INP and thus the formation of new primary ice crystals, it does not display a strong relationship with cloud phase overall. Mignani et al. (2019) show that secondary ice processes are likely the key player in MPCs at a temperature range from -12 °C to -17 °C compared to primary ice formation and droplet shattering. This is further supported by Huang et al. (2021b) who use SOCRATES observations to show that secondary ice processes are important for the ice formation in SLCs. However, depositional growth of ice crystals also accelerates within this regime. A final conclusion regarding the process responsible for the increase in MPC occurrence at this temperature regime could not be drawn based on this data set alone and requires further investigation.

A further comparison of SOCRATES flight observations from D'Alessandro et al. (2021) with our mixed fraction shows a similar distribution across the CTT range in low-level clouds during January and February, however, our mixed fraction shows higher values than the in-cloud flight measurements (Fig. S7). A reason for the seen differences may be that the mixed-phase is underestimated due to a detection limit of small ice particles ($< 50\,\mu m$) by the instruments as discussed by D'Alessandro et al. (2021). Further, their cloud phase is sampled every second which translates to a spatial resolution of roughly 150 m depending on the velocity of the aircraft. In comparison our phase classification has a 1.1 km resolution, thus, about 7 of their cloud phase samples would be observed as one phase in our classification. This could potentially explain the higher number of mixed-phase cases in this study, as D'Alessandro et al. (2021) also show that mixed-phase transects which consist of 20 cloud phase samples are more likely heterogeneous than other phase transects. Thus, phase classifications may well be scale dependent and subject to detection thresholds, which have to be kept in mind when comparing different data sets or evaluating model statistics.

The open to closed fraction for liquid clouds (JJA: 4.54, DJF: 0.51) and SLCs (JJA: 4.40, DJF: 0.57) is similar in the main SO cloud band (50° S–60° S). Thus, this further supports that seasonal differences in cloud phase statistics outweigh any differences found across cloud morphology. Following the hypothesis of preconditioning introduced by Abel et al. (2017) and Tornow et al. (2021), where accelerated transitions from closed to open cells are observed in clouds that formed ice as opposed to SLCs, one may expect to find open MCC clouds to occur more often as MPCs than closed cells. However, we can not observe a higher mixed fraction in the open MCC regime in comparison to closed MCC clouds. Therefore, while preconditioning may impact regional-scale transitions under specific environmental conditions it seems to be only a secondary driver in morphological transitions of marine stratocumuli. However, we can not reliably determine the ice water ratio in our MPCs from spaceborne remote sensing and thus might include MPCs with a very low ice ratio. Eirund et al. (2019a) show that only for a ratio of LWP:IWP (ice water path) of 1:2, the morphological structures of the simulated open cell clouds are impacted by ice formation.

For clouds with cloud-tops below 2.5 km, we find a dependence of the mixed fraction on CTH. This suggests, that deeper, more decoupled boundary layers, where the stratocumulus deck is maintained by detraining cloud cores characterized by larger updrafts, favor ice formation at supercooled temperatures. At the same time, we did not find the mixed fraction to correlate with surface fluxes, which would support the above hypothesis linking the occurrence of convective cloud structures and larger updraft speeds to the increased likelihood of ice formation. Furthermore, the above hypothesis is consistent with modeling

studies that show higher ice occurrence in the updrafts of these clouds (Lee et al., 2021; Yang et al., 2013; Roesler et al., 2017; Young et al., 2018; Eirund et al., 2019b).

The investigation of the link between cloud phase and in-cloud albedo confirms previous results which show that MPCs typically have a higher cloud albedo than liquid clouds (McCoy et al., 2014a, b; Shupe et al., 2006; Achtert et al., 2020). Moreover, the relationship between in-cloud albedo and cloud morphology reveals substantial differences between open and

500 closed cells (0.04 to 0.13) which is consistent with the higher albedo of closed MCC clouds shown by McCoy et al. (2017). These differences in the in-cloud albedo can drive changes in the cloud radiative effect of about $12\,\mathrm{W\,m^{-2}}$ to $39\,\mathrm{W\,m^{-2}}$ depending on season and cloud phase in the SO. We additionally examine the cloud phase difference within the morphological regimes and show that changes in in-cloud albedo across organizational regimes are more pronounced in SLCs than MPCs.

In summary, our results show that seasonal differences in cloud phase for a given CTT are stronger in SO stratocumuli

than organizational changes in cloud phase. Both cloud morphology and phase seem to be primarily constrained by other environmental factors and not by each other. Moreover, this work highlights the importance of improving our understanding of cloud phase and organizational transitions to enhance predictions of cloud albedo in the SO.

*Data availability.* The DARDAR-MASK v2.23 products are available on the Aeris/ICARE data center (http://www.icare.univ-lille1.fr/, last access: December 2020, (Ceccaldi et al., 2013)). MODIS cloud retrievals (MYD06_L2.6) are also obtained from Aeris/ICARE data center

(Platnick et al., 2015).

*Author contributions.* JD performed the data analysis and wrote the article with support of AP. AP established the initial ideas of this study and contributed to its design. OS provided and collocated the remote-sensing retrievals of MODIS on the CALIPSO track. ILM and RW provided the mesoscale-cellular convection classification. All authors contributed to the discussion of the results and editing of the manuscript.

*Competing interests.* The authors declare that they have no conflict of interest.

*Acknowledgements.* The research from JD and AP has been supported by the Federal Ministry of Education and Research (BMBF) "Make our Planet Great Again – German Research Initiative" (grant no. 57429624) implemented by the German Academic Exchange Service (DAAD). Research by ILM is supported by the NOAA Climate and Global Change Postdoctoral Fellowship Program, administered by UCAR's Cooperative Programs for the Advancement of Earth System Science (CPAESS) under award #NA18NWS4620043B. We thank the

520 AERIS/ICARE Data and Services Center for providing access to the data used in this study as well as the CloudSat, CALIPSO and MODIS projects by NASA who provided the original data.

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
