# Peer review of "Exploring Relations between Cloud Morphology, Cloud Phase, and Cloud Radiative Properties in Southern Ocean Stratocumulus Clouds"

_Atmospheric Chemistry and Physics, 2021_

## Author Response (AR1)

**Answered referee's comments**

**Ref 1 Comments:**

Major Comment:

The authors base their analysis on a classification product that was designed to assist implementation of algorithms being developed for the EarthCare satellite. It was adapted to the CloudSat and CALIPSO data sets augmented by thermodynamic information from ECMWF. The authors of this paper have more or less adopted this classification scheme as the primary source of information for their analysis without, in my opinion, sufficient critical assessment of it for their specific purpose. The DARDAR algorithm was developed to be applied globally and was not specifically tuned to the clean maritime environment of the SO where INP concentrations are very low. Therefore, I think the authors should explain in their methods section 1) what actual information, beyond simple model-based temperature threshold relationships, exists at 60m resolution in the vertical profile of an optically thick cloud that is sampled by CloudSat to distinguish phase and 2) how such an algorithm can distinguish or not supercooled liquid drizzle from ice-phase precipitation? Based on Ceccaldi et al. (2013) Figure 1, the DARDAR algorithm uses radar reflectivity and thus the presence of precipitation-sized hydrometeors) along with how that reflectivity is distributed relative to wet-bulb temperature to prescribe phase in optically thick clouds. Thus my third question: 3) To what extent has the DARDAR algorithm been validated in a clean maritime environment where there may not be sufficient INP to nucleate ice phase hydrometeors and where supercooled drizzle may be common in clouds that do not have sufficient updraft strength to initiate secondary ice processes?

We thank the reviewer for their comments and have now integrated a more detailed discussion that addresses points 1 and 2 raised by the reviewer in the revised manuscript.

P5 L138-154: "Furthermore, for temperatures below 0 °C, the radar classification cannot distinguish between supercooled drizzle and ice. In particular in the SO, supercooled drizzle is observed in stratocumulus clouds at temperatures near -10 °C (Mace and Protat, 2018). Further, Silber et al. (2019) show that at the observation station McMurdo, Antarctica supercooled drizzle can persist at temperatures below -25 °C for several hours. While it might be possible that the Mix classification of DARDAR itself is affected as this category is supercooled liquid from lidar and ice from radar. We find it unlikely that multiple layers of Mix could be affected as the lidar would extinguish in the presence of drizzle and the vertical lidar resolution of CALIPSO is 30 m. As most MPCs that contain Mix have mixed layers with a thickness of roughly 480 m (four vertical levels in DARDAR) (see Fig. 1a and S4), the MPCs with identified Mix levels by the radar retrieval are unlikely to be pure drizzle. However, the misclassification of supercooled drizzle as Ice could lead to false identification in MPCs when the lidar is extinguished especially in the cloud category Sup → Ice, as the Ice in these clouds could be supercooled drizzle. Supercooled drizzle is reported to be misclassified as ice by several studies (Cober and Isaac, 2012; Zhang et al., 2017, 2018; Villanueva et al., 2021) in particular at temperatures above -10 °C.

To further test the uncertainties of misclassified supercooled drizzle, we checked how our results are changed if only clouds with an effective radius of 0 µm < $R_e$ < 14 µm at cloud top are investigated. Thus, precipitating clouds should be excluded as $R_e$ > 14 µm at cloud top initiates drizzle (Han et al., 1995; Rangno and Hobbs, 2005; Rosenfeld et al., 2012; Freud and Rosenfeld, 2012). However, we only find slight changes with this threshold (compare Fig.3 with Fig. S2). Further, as MODIS is not able to calculate Re in over 50 % of the identified MPCs (Fig. S1) and as we would also exclude correctly identified precipitating MPCs the threshold of Re is not used as a constraint in this study."

We included a comparison of the evaluation of DARDAR v2 in the Antarctic to address point 3.

P3 L80-83: "While Huang et al. (2021b) report large differences in cloud phase detection between various satellite products, which struggle specifically with MPCs, they use the DARDAR v1 which is known to overestimate supercooled liquid. In contrast, DARDAR v2 is validated with several ground-based measurements in the Antarctic by Listowski et al. (2019) who also show that DARDAR v2 has the ability to capture the seasonal cycle of supercooled liquid cloud fraction. "

However, we do agree with the reviewer that some clouds with low ice crystal number concentrations may not be identified as mixed phase. This limitation is now explicitly addressed and discussed in the revised manuscript:

P3 L84-85: "Nevertheless, MPCs with very low ice crystal number concentrations which are common in the SO might still be misidentified as supercooled liquid."

Specific Questions Related to this Comment:

Line 109: This logic to derive a column phase type seems reasonable. However, the authors must explain what in the DARDAR algorithm distinguishes the liquid types from the mixed types? What, in the actual data, is providing the information? If it is the wet-bulb temperature threshold combined with radar reflectivity, then it is important to explain whether this threshold can identify the presence of liquid supercooled drizzle in optically thick closed cell stratocumulus.

Please refer to the answers given to points 1 and 2 above.

Line 115: Supercooled drizzle has been observed in Southern Ocean stratocumulus clouds to temperatures of -25C. It seems as though the wet bulb threshold might have difficulty identifying the presence of supercooled drizzle. Mace and Protat (2018 DOI: 10.1175/JAMC-D-17-0194.1) document supercooled drizzle occurrence with a cloud temperature near -10C. See also Silber, I., Fridlind, A. M., Verlinde, J., Ackerman, A. S., Chen, Y.-S., Bromwich, D. H., et al. (2019). Persistent supercooled drizzle at temperatures below −25 observed at McMurdo Station, Antarctica. Journal of Geophysical Research: Atmospheres, 124, 10878–10895. https://doi.org/10.1029/2019JD030882

Please refer to the answers given to points 1 and 2 above.

Line 151: Again, what "signals"? Where is the information coming from? From Mace et al. (2021) CALIPSO is mostly unable to identify the presence of ice when it occurs in optically thick clouds.

Please refer to the answers given to points 1 and 2 above, and we adjusted the sentence as follows:

P8 L192: "While most of our MPCs (more than 95 %) contain a Mix layer that is determined by both the radar (ice) and the lidar (supercooled liquid), we also include Sup over Ice clouds in our MPC classification. "

Line 160: The value of 0.5% is 20 times lower than 10% that is reported by Mulmenstadt. Mace et al. (2021) analyzed cloudsat and calipso data between 2007 and 2010 and find that ~25% of the SO MBL clouds are precipitating. I am aware that Marchand and his student (paper in review) find a substantially larger precipitating fraction from data collected at Macquarie Island. The magnitude of this discrepancy between what is found in this paper and what has been reported in the past is large and those differences have implications for our understanding of Southern Ocean climate.

Thank you for raising this issue which lead us to revise our diagnostic and further discuss precipitation. We now clarified more directly that we are not trusting DARDAR with precipitation and include new values of precipitating clouds which we calculated by the percentage of clouds with $R_e$ > 14μm. These values of precipitating clouds are in agreement with previous studies.

P8 L199-219: "Of our liquid low-level clouds, about 90 % in austral winter and 60 % in summer are supercooled at cloud top and almost all of them (99 %) belong to the Sup category and thus remain supercooled at cloud base in the SO. We identify almost no low-level liquid clouds that show DARDAR rain categories below cloud base (~ 0.5 %). However, the DARDAR algorithm was not primarily designed to detect precipitation and as the CloudSat radar is contaminated by surface clutter, only heavy and moderate drizzle can be detected at heights below roughly 720 m and 860 m, respectively (Marchand et al., 2008). Thus, for many liquid low-level clouds which have a CTH of around 1.2 km (Fig. 2 g–h) light drizzle rates at cloud base could have been missed at lidar saturation which explains the too low drizzle rates. If we define precipitating clouds as clouds with an effective radius of $R_e$ > 14 μm then roughly 10 % and 3 % of low-level liquid clouds are precipitating in winter and summer, respectively (Fig. S1 a and c). These values are in agreement with Mülmenstädt et al. (2015) who show that the rain probability of liquid clouds at all levels is roughly 10 % at 45° S and 3 % at 60° S. Though they use DARDAR v1 to identify the cloud phase, they use the 2C-PRECIP-COLUMN product based on retrievals from CloudSat to calculate rain probability. Further, this latitudinal gradient in precipitation is also reported by Mace et al. (2021) who investigate MPCs with satellite and ground-based measurements. They also show that about 33 % of MPCs in the SO produce supercooled precipitation. We find similar values of 30 % (winter) and 40 % (summer) of MPCs that have Re > 14 μm (Fig. S1 a and c). Additionally, we find that on average liquid low-level clouds are 57 % optically thinner than their mixed-phase counterparts calculated independent of season and thus are more unlikely to contain sufficient water content to generate precipitation. As most liquid clouds are optically thin and are not precipitating especially in summer, this could

either hint towards a mixed-phase detection bias in DARDAR, which we find unlikely as discussed above or suggest that most optically thicker supercooled liquid clouds generate ice and become MPCs. Interestingly, liquid (and supercooled liquid) closed MCC clouds are optically thicker than open and low-level clouds. This could indicate a potential link between cloud phase and cloud morphology, however, as discussed below we find no further evidence for this link."

Line 165: Might it also be that optically thin clouds are much more likely to be categorized as liquid because they are optically thin and that optically thick layers are more likely to be classified as mixed because it is impossible to distinguish the phase of the precipitation that is in them?

Please refer to the answers given to points 1 and 2 above. Further, we explicitly checked for this and the overall result did not change for $0\mu m < R_e < 14\mu m$.

Line 419: Might it be that my third question above has been addressed by the authors' comparison to the in situ data analysis of D'Alessandro et al. (2021) where the aircraft data show a lower occurrence of MPC than what is suggested in the DARDAR product? I wonder if the authors should reconsider the possibility that the lower occurrence of MPC in the aircraft data may actually be indicative of the inability of the radar reflectivity-wet bulb relationship to distinguish the difference between supercooled liquid precipitation and ice-phase precipitation?

Thank you, we now also tested if our comparison changes for clouds with $0 \ \mu m < R_e < 14 \ \mu m$ (See Fig.S7). However, as this does not influence the results, we are confident that some amounts of ice are likely present in the Mix category or that if it is drizzle it formed via the ice phase, which means the cloud had been impacted by ice-phase processes.

P14 L331-333: "Further, a direct comparison of SOCRATES flight observations from D'Alessandro et al. (2021) with our mixed fraction in Fig. S7 shows a similar trend across the CTT range in low-level clouds during January and February, though our mixed fraction shows higher values than the in-cloud flight measurements."

P19 L468-478: "A further comparison of SOCRATES flight observations from D'Alessandro et al. (2021) with our mixed fraction shows a similar distribution across the CTT range in low-level clouds during January and February, however, our mixed fraction shows higher values than the in-cloud flight measurements (Fig. S7). A reason for the seen differences may be that the mixed-phase is underestimated due to a detection limit of small ice particles (< 50 μm) by the instruments as discussed by D'Alessandro et al. (2021). Further, their cloud phase is sampled every second which translates to a spatial resolution of roughly 150 m depending on the velocity of the aircraft. In comparison our phase classification has a 1.1 km resolution, thus, about 7 of their cloud phase samples would be observed as one phase in our classification. This could potentially explain the higher number of mixed-phase cases in this study, as D'Alessandro et al. (2021) also show that mixed-phase transects which consist of 20 cloud phase samples are more likely heterogeneous than other phase transects. Thus, phase classifications may well be scale dependent and subject to detection thresholds, which have to be kept in mind when comparing different data sets or evaluating model statistics."

Minor Comments:

Line 87: The asymmetry parameter g is a function of the droplet size distribution (effective radius and width of the distribution). How might g vary in realistic southern ocean clouds?

Thank you, we agree that we should have discussed this. Therefore, we provided additional information in a new paragraph.

P4 L97-103: "McFarquhar and Cober (2004) find that MPCs peak at g = 0.85 and liquid clouds at g = 0.87. Further, Gayet et al. (2002) show that in MPCs the asymmetry parameter ranges from 0.82 to 0.85 which is similar to values in liquid clouds. They find higher values of g are typically found in liquid clouds with high liquid water content whereas lower values of g (0.73–0.80) are found in ice clouds. This corresponds to findings by Shcherbakov et al. (2005) and Xu et al. (2022) who demonstrate that the asymmetry parameter is g = 0.77 in cirrus clouds in the SH. As differences between liquid clouds and MPCs are similar the asymmetry parameter g = 0.85 is used for both liquid and MPCs"

Line 101: Since CloudSat's reported resolution is 240 m, it seems that there may not be much difference in what is known about phase or anything else between 720 and 780 m in an optically thick cloud. Also, the conservative rule of thumb is that clutter begins at 1 km. However, this could be checked by examining the CloudSat cloud mask product to see if the Marchand et al., algorithm is identifying clutter or not at a particular height since the height where clutter begins varies by 1-2 (240 m) range bins around an orbit.

To address this comment we included a new paragraph on ground-based clutter.

P4 L116-123: "The surface cluttering of the radar can cause noise up to 2 km which can not be clearly distinguished from the signal, especially at heights below 720 m and thus clouds are missed (Marchand et al., 2008). Even though some studies (Liu et al., 2012; Fletcher et al., 2016b) consider anything roughly below 1 km as ground clutter, Mioche et al. (2015) show that in comparison with ground-based observation the cloud fraction of DARDAR is 10 % lower from 500 m to 1000 m while in the range from 0 m to 500 m it is 25 % lower. Thus, in this study, we consider 720 m as the threshold for surface clutter similar to other studies (Kay and Gettelman, 2009; Huang et al., 2017; Noh et al., 2019; Listowski et al., 2019). In order to correctly identify the cloud phase, however, we require one level below the liquid CBH. Thus, we restrict to only clouds with a liquid CBH at 780 m or above."

Regarding the comment on differences between 780m versus 720m, we agree that both levels are within the radar resolution and thus the radar can not provide further information. However, as the lidar resolution is 30m we rely on the lidar showing one more level as clear below the liquid cloud base. Thus, close to the surface (<1km) optically thick clouds may be missed as the liquid cloud base may be at heights below 780m and the lidar is not yet extinguished at these heights in contrast to clouds with CTH above 1km. This is briefly discussed in the results section:

P16 L361-363: "As clouds with CTHs below 1 km can only have a small vertical extent, this could potentially lead to a bias towards SLC occurrence at CTHs below 1 km as thicker clouds tend to form ice as discussed in Sect. 3.1."

Line 104: I'm not sure I understand "orginal" in this context. The actual vertical resolution of CloudSat is 480 m and it is oversampled twice to provide return in 240 m range bins. The 60 m resolution of DARDAR is purely a construct and represents an extrapolation of the CloudSat data. That vertical resolution is a function of the CALIPSO lidar data. Information at 60 m resolution below the point where the lidar attenuates is likely not a function of the actual data but may be due entirely to the model-derived therodynamics and the DARDAR decision tree.

Thank you for raising this point. We completely agree and have reworded the paragraph as such:

P5 L124-126: "As the constructed vertical resolution of DARDAR is 60 m, three levels equal a distance of 240 m which is also the oversampled vertical resolution of CloudSat (effective vertical resolution 480 m). Thus, this distance ensures that multi-layer clouds are two separated clouds with a sufficiently large separation."

Line 121: Problems with the MODIS CTT are a potentially important piece of information for the community. Can the authors elaborate, provide an example, or provide a reference if this problem has been previously reported?

We added the following paragraph to the revised manuscript and the issue itself is documented in Fig. S5.

[Figure]

P6 L160-162: "From a brief visual inspection, it seems to be related to jumps in MODIS CTH which are not detected by the active satellites. Further, we find that the MODIS CTH is often higher than that of DARDAR."

Line 156: What does "internally mixed" mean in this context? If it means that the DARDAR algorithm is identifying the presence of ice and liquid, then, again, I ask where the information is coming from and can it distinguish supercooled liquid precipitation from ice-phase precipitation?

Please refer to the answers given to points 1 and 2 above.

P5 L141-145: "While it might be possible that the Mix classification of DARDAR itself is affected as this category is supercooled liquid from lidar and ice from radar. We find it unlikely that multiple layers of Mix could be affected as the lidar would extinguish in the presence of drizzle and the vertical lidar resolution of CALIPSO is 30 m. As most MPCs that contain Mix have mixed layers with a thickness of roughly 480 m (four vertical levels in DARDAR) (see Fig. 1a and S4), the MPCs with identified Mix levels by the radar retrieval are unlikely to be pure drizzle."

line 445: Please define in-cloud albedo. Apologies if I missed this definition earlier.

Thank you, the in-cloud albedo is defined at line 93.

**Ref 2 Comments:**

The phase product is performed as a vertical integration. As I understand it, any combination of liquid/ice/mixed pixels will result in the classification of the column as mixed-phase (as long as there is at least one pixel containing liquid and/or ice). Can you comment on this in relation to potentially restricting liq<<(liq+ice) from being classified as mixed phase assuming both liq and ice are greater than 0? (e.g., for in situ observations, often mixed-phase is classified as 0.9>LWC/TWC>0.1, such as described in Korolev et al. (2017) mixed phase review paper).

Even the phase retrieval itself is associated with biases discussed in the methods section. These are further confounded by retrievals of ice and liquid water paths in mixed-phase cloud scenes. These retrievals are underconstrained by direct observation and assumptions have to be made that can substantially impact liquid water content and ice water content diagnostics. In this study, we attempted to evaluate a multiannual dataset for the entire SO with as few uncertainties as possible. However, this conservative approach is not faultless either and our conclusions may well be impacted by this. For instance, Eirund et al. 2019 show that only for a ratio of LWP: IWP of 1:2 cloud morphology was impacted.

We include a discussion of this point in the revised version of the manuscript as one of the potential limitations of this study.

P3 L84-85: "Nevertheless, MPCs with very low ice crystal number concentrations which are common in the SO might still be misidentified as supercooled liquid."

P19 L486-489: "However, we can not reliably determine the ice water ratio in our MPCs from spaceborne remote sensing and thus might include MPCs with a very low ice ratio. Eirund et al. (2019a) show that only for a ratio of LWP:IWP (ice water path) of 1:2, the morphological structures of the simulated open cell clouds are impacted by ice formation."

You refer to albedo as reflectivity numerous times in the paper which can be confusing (especially since it's a remote sensing paper). It might be best to just say albedo.

Thank you, we agree and adjusted it.

Line 19: "These differences in cloud albedo"*

We corrected it, thanks.

Line 55-57: This sentence is confusing

To improve clarity we rewrote this sentence as:

P3 L55-57: "Eirund et al. (2019a) demonstrate in a case study of Arctic stratocumulus that mixed-phase open MCC clouds have a larger cell size than pure liquid open cells"

Figure 2: Change y-axis label to just "Probability"

We agree and adapted it.

Figure 2 caption: Can you specify what category you are referring to? Are they normalized by cloud regime type and respective row variable? Then I don't think you need to add the category comment.

 You are right we now specified it and the caption now states:

 P9  Figure 2. Seasonal probability density functions (PDFs) of (a–c) CTT, (d–f) COT, (g–i) CTH, (j–l) latitude and (m–o) cloud albedo for (left) open MCC, (middle) closed MCC, and (right) low-level clouds with bin width of 1 °C, 1, 120 m, 1° and 0.05, respectively. The PDFs are normalized for each cloud regime type, phase and season individually. In JJA only 5.1 % of the annual closed MCC clouds occur and therefore closed MCC in JJA are indicated by more transparent color shading.

Line 166: Where did you get 55% from? Did you calculate it by using all the COT values (including different seasons and cloud regime types) in table 1? If so, please specify.

We now specified that this is for all low-level clouds for all seasons.

P8 L212-214: "Additionally, we find that on average liquid low-level clouds are 57 % optically thinner than their mixed-phase counterparts calculated independent of season and thus are more unlikely to contain sufficient water content to generate precipitation."

Line 230: Introduce SLCs here (and not at line 238). Also, it's slightly confusing to introduce this since you introduce SLC (the same acronym except clouds isn't plural) at line 10.

Thank you for noticing it was not intentional and we adjusted it.

Line 258-259: Why isn't the peak at -5C observed for closed cell MCC?

Thank you for raising this issue, we now added the following information.

P12 L304-307: "Surprisingly, this peak is not observed in closed MCC clouds. This could potentially be related to a detection bias close to 0 °C as the mixed fraction at temperatures above -10 °C is lower for clouds with 0 µm < $R_e$ < 14 µm (Fig. S2). However, even for clouds with 0 µm < $R_e$ < 14 µm this secondary peak is barely detectable and much weaker than in open MCC and all low-level clouds."

Line 342-343: What part of your results suggest mixing is relevant to CTH? You just mentioned data limitations prevent you from evaluating turbulence/circulations and no significant trends were found between SST and sfc wind speed.

We agree, that we cannot prove this from our results. Therefore, we rephrased the sentence even more carefully.

P 17 L391: "Overall, this could suggest that the mechanisms of mixing (turbulence and circulation) may play a larger role in CTH than previously thought (McCoy et al., 2017)."

**Ref 3 Comments:**

L47 - Are favored?

Thank you, the sentence was reworded as:

P2 L47-48: "McCoy et al. (2017) show that open MCC clouds preferentially form during marine cold air outbreaks."

L78 - Some previous studies have only used DARDAR phase at the cloud top. How reliable is the phase deeper into the cloud where the lidar has attenuated (or does this not matter)?

Though some studies only use DARDAR at cloud top, many studies rely on the whole vertical cloud mask (e.g. , Mason et al. 2014, Fletcher et al. 2016 and Listowski et al. 2019). To further ensure that the mixed clouds are really MPC, we tested to add a threshold for effective radius between 0µm < $R_e$ < 14µm to exclude supercooled drizzling clouds which did not affect our results. Please also refer to point 1 and 2 from Ref 1.

L89 - does g change for ice or is the same value used for all clouds? The connection to optical depth would also presumably depend on the value assumed in the retrieval.

Thank you, we agree that we should have discussed this. Therefore, we provided additional information in a new paragraph.

P4 L97-103: "McFarquhar and Cober (2004) find that MPCs peak at g = 0.85 and liquid clouds at g = 0.87. Further, Gayet et al. (2002) show that in MPCs the

asymmetry parameter ranges from 0.82 to 0.85 which is similar to values in liquid clouds. They find higher values of g are typically found in liquid clouds with high liquid water content whereas lower values of g (0.73–0.80) are found in ice clouds. This corresponds to findings by Shcherbakov et al. (2005) and Xu et al. (2022) who demonstrate that the asymmetry parameter is g = 0.77 in cirrus clouds in the SH. As differences between liquid clouds and MPCs are similar the asymmetry parameter g = 0.85 is used for both liquid and MPCs"

L95 - What is mix? My understanding is that it is a lidar backscatter peak along with a radar return. Is it possible that this is not actually mixed phase cloud, but perhaps precipitation (or just large liquid water droplets)?

Thank you for raising this issue as well, to further ensure that the supercooled above ice clouds are really MPC we tested to add a threshold for effective radius which needs to be between 0μm < $R_e$ < 14μm. This prevents to identify supercooled drizzle as ice. Please also refer to point 1 and 2 from Ref 1.

L107 - Presumably Ice only is also a vertical 'phase' that occurs - or is that excluded?

Thanks, yes pure ice clouds are excluded because the MCC algorithm is based on LWP. For clarity, we state this explicitly in the methods section.

P4 L113-115: "The height of the highest cloud level is defined as the cloud top height (CTH) and the lowest as the liquid cloud base height (CBH). Thereby excluding pure ice clouds which we exclude as the MCC algorithm is based on the LWP (see Sect. 2.3). "

L121 - Is there an example of this? I would have thought the MODIS CTT might be a better option, as at least for these low clouds, MODIS is actually observing the temperature, rather than reconstructing it from the p-T relationship in ERA5? Is the uncertainty perhaps due to cloud phase errors?

We added the following paragraph to the revised manuscript and the issue itself is documented in Fig. S5.

[Figure]

P6 L156-163: "The ECMWF cloud top temperature (CTT) is defined as the temperature from ECMWF at CTH. As shown in four examples in Fig. S5, our data

set, which is combined with MODIS, also provides the CTT from MODIS. However, we decide to use the ECMWF CTT for two reasons: 1) because it will be more consistent with the DARDAR classification methodology which is also based on the ECMWF temperature and further because CTH between DARDAR and MODIS varies and 2) because the MODIS CTT exhibits unrealistically large and abrupt changes of more than 10 °C within a distance of 2 km (Fig. S5). From a brief visual inspection, it seems to be related to jumps in MODIS CTH which are not detected by the active satellites. Further, we find that the MODIS CTH is often higher than that of DARDAR."

L165 - I am not sure what is going on here. The paragraph suggests that Mulmenstadt et al find more precipitating clouds the the current study, but also that they rarely find precipitating clouds (although more often than this study). Is it clear why these studies disagree given they both use very similar data)?

We agree, it was a bit confusing. The main reason they disagree on rain probability is that Mülmenstadt uses a different product for rain probability than we do. Though we both use DARDAR for cloud phase. We rephrased this paragraph and also extend the discussion to precipitating clouds with $R_e > 14\mu m$.

P8 L199-219: "Of our liquid low-level clouds, about 90 % in austral winter and 60 % in summer are supercooled at cloud top and almost all of them (99 %) belong to the Sup category and thus remain supercooled at cloud base in the SO. We identify almost no low-level liquid clouds that show DARDAR rain categories below cloud base (~ 0.5 %). However, the DARDAR algorithm was not primarily designed to detect precipitation and as the CloudSat radar is contaminated by surface clutter, only heavy and moderate drizzle can be detected at heights below roughly 720 m and 860 m, respectively (Marchand et al., 2008). Thus, for many liquid low-level clouds which have a CTH of around 1.2 km (Fig. 2 g–h) light drizzle rates at cloud base could have been missed at lidar saturation which explains the too low drizzle rates. If we define precipitating clouds as clouds with an effective radius of $R_e > 14$ µm then roughly 10 % and 3 % of low-level liquid clouds are precipitating in winter and summer, respectively (Fig. S1 a and c). These values are in agreement with Mülmenstädt et al. (2015) who show that the rain probability of liquid clouds at all levels is roughly 10 % at 45° S and 3 % at 60° S. Though they use DARDAR v1 to identify the cloud phase, they use the 2C-PRECIP-COLUMN product based on retrievals from CloudSat to calculate rain probability. Further, this latitudinal gradient in precipitation is also reported by Mace et al. (2021) who investigate MPCs with satellite and ground-based measurements. They also show that about 33 % of MPCs in the SO produce supercooled precipitation. We find similar values of 30 % (winter) and 40 % (summer) of MPCs that have $R_e > 14$ µm (Fig. S1 a and c). Additionally, we find that on average liquid low-level clouds are 57 % optically thinner than their mixed-phase counterparts calculated independent of season and thus are more unlikely to contain sufficient water content to generate precipitation. As most liquid clouds are optically thin and are not precipitating especially in summer, this could either hint towards a mixed-phase detection bias in DARDAR, which we find unlikely as discussed above or suggest that most optically thicker supercooled liquid clouds generate ice and become MPCs. Interestingly, liquid (and supercooled liquid) closed MCC clouds are optically thicker than open and low-level clouds. This could indicate a potential link between cloud phase and cloud morphology, however, as discussed below we find no further evidence for this link."

Fig. 3 - When the circles overlap (particularly on the top right), it can be difficult to see how they change. It also makes it difficult to determine how the mean phase fraction changes too, as the central points are black for both mixed and liquid.

Thank you, we now increased the size of the central points to better show their different colors.

L254 - 'could be related to' - you have the data to show if this is the case I think? This paragraph jumps about a bit between potential explanations and new results, it might be easier re-ordered slightly (although I leave that to the authors discretion).

We agree and rephrased this sentence. Thank you for your comment, but we decided against changing the order as in our opinion though it jumps between potential explanations and new results they build on each other.

L285 - the decrease in mixed fraction for the observed mixed fraction - this sentence is a bit confusing. It might be useful to be more explicit about which direction the mixed phase fraction is changing with temperature (and which bit you are considering here).

To improve the clarity of this sentence we have rewritten it to:

P14 L 335-337: "On the other hand, we find it unlikely that CTH-dependent INP limitation is the primary cause for the decrease in mixed fraction below -16 °C at high CTHs as it seems to be unaffected by CTH (Fig. S4)."

Fig. 4 - L291 states that there is not a higher mixed phase fraction in open than closed MCC clouds. This figure appears to demonstrate that the opposite is the case, and that there is a considerably higher mixed phase fraction in closed MCC, once latitude and CTT are accounted for (if I am reading it correctly)? Is it possible that Fig. 3 shows little difference because it is not stratifying by the correct variables?

Thank you for this comment, we agree that this sentence with the not is confusing and agree with you that the mixed fraction in closed MCC is higher also in Fig. 3 especially if you look at DJF in agreement with Fig. 4. Thus, we have rewritten the sentence from the old version to the new one in the revised manuscript.

Old P13 L291: "In agreement with the findings of Sect. 3.1, there is not a higher mixed fraction in open than closed MCC clouds."

New P15 L341: "In agreement with the findings of Sect. 3.1, the mixed fraction of closed MCC clouds is higher than that of open MCC clouds."

Fig. 5 - Given you need several levels to identify a liquid phase top, but the cloud must have a CBH above 780m. Similarly, presumably at least one cloudsat layer outside the clutter is required - does that also set a minimum useful CTH? Might this contribute to the sharp shift in cloud phase for the thinnest clouds (CTH <1km)?

We agree we look at very thin clouds in that range by design of our retrievals. However, as the trend continues until >2km in depth we argue that it suggests that there may be a relationship here. Further, even if the thin clouds around 1km did

generate some ice they would likely be missed by the radar (ice being too small or too low in concentration) and thus not classified as such.
We decided not to set a minimum CBH as we still find differences between the different regimes and also discuss that this sharp increase is probably due to limited CBH.

P16 L361-364: "As clouds with CTHs below 1 km can only have a small vertical extent, this could potentially lead to a bias towards SLC occurrence at CTHs below 1 km as thicker clouds tend to form ice as discussed in Sect. 3.1. However, this is the same for all seasons and cloud morphologies. Thus, differences across seasons and between open and closed cells can still be interpreted. "

L371 - How much of this is due to different sampling of 'cloudy' pixels with the MODIS algorithm? Are enough cloud edge pixels discarded in the open cell regime to make a difference here (as presumably almost all closed cell pixels have a valid retrieval?)

We only use pixels that are identified as cloudy by MODIS for this. Thus retrievals of partly cloudy pixels do not impact our statistics.